# Performance Analysis of an Open-Flow Photovoltaic/Thermal (PV/T) Solar Collector with Using a Different Fins Shapes

**Mohammed G. Ajel [1], Engin Gedik [2]**, **Hasanain A. Abdul Wahhab [3],\*** **and Basam A. Shallal [1]**

1. Institute of Graduate Programs, Department of Energy Systems Engineering, Karabük University, 78050 Karabuk, Turkey
2. Faculty of Technology, Department of Energy Systems Engineering, Karabük University, 78050 Karabuk, Turkey
3. Training and Workshop Center, University of Technology-Iraq, Ministry of Higher Education and Scientific Research, Baghdad 10066, Iraq
* Correspondence: 20085@uotechnology.edu.iq; Tel.: +964-781-130-9446

**Abstract:** Generally, photovoltaic panels convert solar energy into electricity using semiconductor materials in their manufacture by converting energy into electricity by absorbing heat from solar radiation, which requires reducing the heat of these panels to improve the efficiency of electricity generation. Therefore, the issue of cooling photovoltaic panels became one of the objectives that were addressed in many studies, while cost reduction was the most important concern in the manufacture of these panels, followed by low energy consumption. In this work, the performance analysis for PV panels was achieved through using two models (Model-C and Model-S) of open-flow flat collector improves the cooling process for PV panel. The investigations of open-flow flat collector have been performed and analyzed using experimental and numerical methods. The simulation analysis was carried out by ANSYS FLUENT 17.0 software with two open-flow flat collector modules. Results appeared the effect of collector design (fin shape) on PV/T system performance and PV panel temperature, it was the percentage of difference temperature with uncooled PV panel 8.4% and 9.8% for Model-C and Model-S, at 1:00 p.m., while the performance of PV panel increased to 23.9% and 25.3% with both models, respectively at (1:00 p.m.). The evaluation result demonstrates that the performance of PV/T system increased, also the fins in open-flow collector helped the system enhance.

**Keywords:** collector performance; collector thermal efficiency; PV/T system thermal controller; water open flow collector

## 1. Introduction

Flat plat collector system (FPCS) is commonly used in building sector where low and medium operating temperatures are required for domestic water heating or used for cooling higher temperature systems [1,2]. Among evolving technologies of harvesting solar energy into more useful forms of energy are the solar collectors [1,3]. Prominent among them is the FPSC [4]. The FPSC are widely used for solar water heater (SWH) and space heating in buildings where low and medium operating temperature is considered for domestic hot water production [5,6]. Othman et al. [7], to quantify the thermal and electrical efficiency, conducted theoretical and practical investigations of the influence of the fins on the solar hybrid air collector were conducted. Using air for heat transfer, for solar cells, and to boost electrical efficiency, thus lowering the operating temperature, will help achieve a decent level. It was determined that cooling medium, such as fins, are required. The fins are a crucial aspect of the development of absorbent elements to accomplish the PV/T hybrid's high-power thermal and electrical efficiency [6–8]. In Jin et al. [9], it has been found to be suitable for warm temperature applications by PV panel system. The PVT system helps decrease collector heat loss. When solar cells operate as a heat absorber, and when a windshield is added, heat loss is reduced even further, but reflecting losses rise. A new design is being researched

and developed. Water collector in the form of a flat box for polycrystalline PV modules. Aluminum alloy is used to make and test the heat sink. The findings concerned strength performance. He discovered a thermal efficiency of around 40%, which is roughly 0.8 for a traditional solar heat collection device. The new energy system has a better efficiency than the previous system [10–13]. The AL-Shamani et al. [14] simulation revealed that numerous design elements and operational circumstances influence the effectiveness of hybrid PVT. As a result, seven of the combinations are tailored to sorption complexes. A well-absorbed design with great efficiency was also evaluated, contrasted, and simulated (total efficiency). The mathematical system of a variety characteristics was examined including: solar radiation, flow rate, and air temperature, and decided that the collector should be a heat collector with a flat plate and a glass plate. Through mathematical research, it was discovered after experimented on several systems at same conditions. The helical flow design was found to be the most efficient, with a maximum thermal efficiency of 50.12 percent and an electrical cell efficiency of 11.98 percent. In Hussein et al. [15], two types of thermal photovoltaic PV/T systems were constructed and tested in the Singapore climate. The first type of PV modules feature monocrystalline Si cells and employ a paper tube type thermal collector, whereas the second type PV modules do not. The photovoltaic module is made up of polycrystalline solar cells and a parallel thermal collector. Experiments were carried out under everyday settings at mass flow rates ranging from 0.03 kg/s to 0.06 kg/s. According to the results, the average thermal and photoelectric effectiveness of first type is 40.7 percent and 11.8 percent, respectively, while second type is 39.4 percent and 11.5 percent, respectively. The average photovoltaic effectiveness of PVT modules is found to be around 0.4 percent higher than that of a standard PV module [16–22]. Abdul-Ganiyu et al. [19] designed and tested of solar roofing systems to enhance electricity efficiency and supplied on-site hot water applications in their study. Their systems are made up of an ordered succession of amorphous solar cells with an oscillating flux linked by glass wool from the bottom of the photovoltaic panel [23,24]. With a combined efficiency of 70.53 percent to 81.5 percent, respectively, the PV/T collector is fitted for a portable solar tracker that can be recorded total data for example, mass flow rate and signed to expose to the greatest amount of radiation with changing variables, such as mass flow rate and solar radiation [25–28].

Hybrid solar PV/T generates electricity and heats air and water. All previous research studies propose a new PV cooling have demonstrated that researchers are eager to find a way to harness solar energy. In this paper, a new design of PV/T cooling collector has been studied experimentally and numerically. The main objective of this study is to investigate the thermal and electric efficiency of newly designed PV/T systems under the various parameters experimentally and numerically. For this purpose, an experimental test rig was designed and built then a series of experiments were conducted. Additionally, the computationally simulated transient thermal behavior of the PV/T cooling collector system was accomplished using ANSYS fluent 17.0. Obtained results were depicted graphically and discussed in detail.

## 2. Materials and Methods

This section describes the main research strategy and the techniques used in the study, namely experimental investigation and CFD simulation. Two experimental open-flow flat collector models were designed and built to confirm prediction and CFD simulation results. The models were subjected in relation to a measurement program, which will be described further in the section. The entire experimental facility is discussed the changing the design of collector and how does it affect the cooling of the photovoltaic panel (PV), describe of open-flow flat collector analysis utilized, measurement system concerning the characterization of collector performance, and uncertainty analysis.

*2.1. Experimental Facility*

2.1.1. Layout of the PV/T System

The PV/T system performance is influenced by the working environment. The open-flow flat collector models developed and built in this study have been installed at the University of Technology in Iraq's solar research site. The latitude is N 33.3123°, the longitude is 44.446 E, and the elevation is 21.23 m. Meteorological data measured within the university for sunrise (7 a.m.–7 p.m.) as a working period of the open-flow flat collector are as follows: the average wind speed is 1.7 m/s; the intensity range of solar radiation is 24 to 1112 W/m$^2$; and the average ambient air temperature (and range) is 34 °C (26–43 °C). The open-flow flat collector models are divided into two categories: Model-C and Model-S.

2.1.2. Design of Open-Flow Flat Collector

The temperature drops from the amount of solar radiation reaching the PV panel's back surface determines the flat collector surface. The area of the collector, and the absorber materials. As the future goal is to increase the amount of electricity generated by the PV panel by using open-flow flat collector, it is suggested that it be installed in an open environment and use water as a solar radiation absorbing medium. As a result, the flowing water would be the absorber material, while a collector cover with high conductivity could be used; preferably for investigation objectives, aluminum plates would be used. The collector area could be calculated depending on the area of PV panel and heat flux which reaches this cover in the model location. The collector has a slope angle 30° to enhance the projection model of solar radiation within its daily range.

The function of the open-flow flat collector is to subtract the heat from the back surface of the photovoltaic panel to the water flowing into it. The parameters involved in the collector are the tangential and vertical parts of internal geometry, and the increasing of contact area between the heat transfer material and hot surfaces; furthermore, increasing the temperature difference between water entering FPC and ambient air. The new design utilizes aluminum uniform bulges connected with collector cover from bottom to provide powerful heat transfer and water movement in a smooth, slow motion. This is possible because the velocity components of water flow are more value when the collector cover was without bulges. In the new design of the present work, two models of bulges utilized: cubic shape (Model-C), and sphere shape (Model-S). The FPC geometry for these models could be described as follows:

Design and fabricate Model-C: this model has a number 120 cubic bulges, dimensions 15 × 15 × 15 mm, and is made of aluminum to allow powerful heat transfer. These bulges were evenly distributed on the collector cover's bottom surface so that they were eight cubes within one row, while the number of rows was 12, as shown in Figure 1a.

Design and fabricate Model-S: this model has a number 120 sphere bulges, a 25 mm diameter, and is made of aluminum to allow powerful heat transfer. These bulges were evenly distributed on the collector cover's bottom surface, so they were eight spheres within one row, while the number of rows was 12, as shown in Figure 1b.

While the collector body has dimensions of 600 × 1100 × 30 mm, a 13 mm inlet and outlet diameter, it is made of steel. To connecting collector cover was used number of screws and rubber band using to prevent water leakage from the edges of the collector.

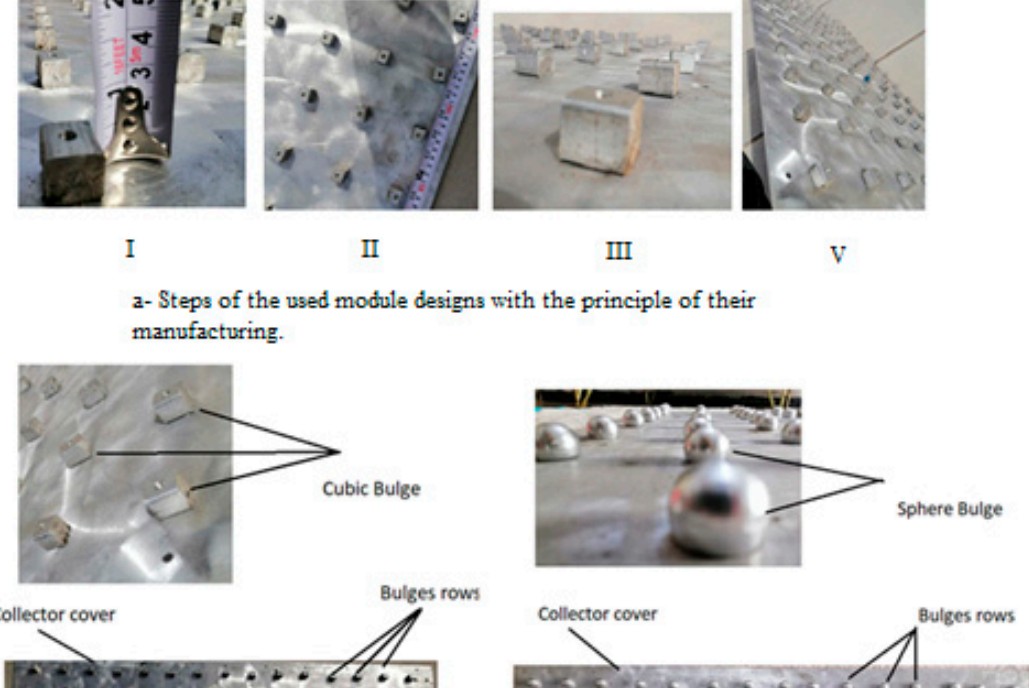

**Figure 1.** Structural pictures of collector models; (**a**) steps of the used module designs, (**b**) model-C, and (**c**) model-S.

### 2.1.3. Experimental Setup

Experimental investigation of PV panel cooling system design for power generation is an objective of present study. The subtract heat energy was done using open-flow flat collector by depend on closed circle water flow. The thermal collector (absorber) is mounted on the back of a standard PV panel. To obtain an initial estimate of the characteristics of the PV/T system, all experiments were carried out in comparison with the PV system, and a portion of the collected energy was extracted as electricity rather than heat, as shown in Figure 2. Both the PV panel and PV/T system were inclined 30° from the horizontal plane. Temperatures for different regions of PV/T system were measured by temperature measurement unit, and the water flow rate was measured by flow rate device before entering the collector. A one-way valve was used to precisely control the water flow rate. Experimental procedures were carried out in sunny days and avoiding cloudy days or clouds during experiments. All tests of temperature at different positions of open-flow flat collector model were carried out to be familiarized and confident with the measuring procedure. Experimental measurements were realized at the solar research site where one of the two PV panels includes the open-flow flat model installed. Temperature measurement can be divided into three groups; first group was used three thermocouples for measuring PV panel surface, second group was used two thermocouples for measuring inlet and outlet water temperature, and third group was used three thermocouples for measuring collector surface.

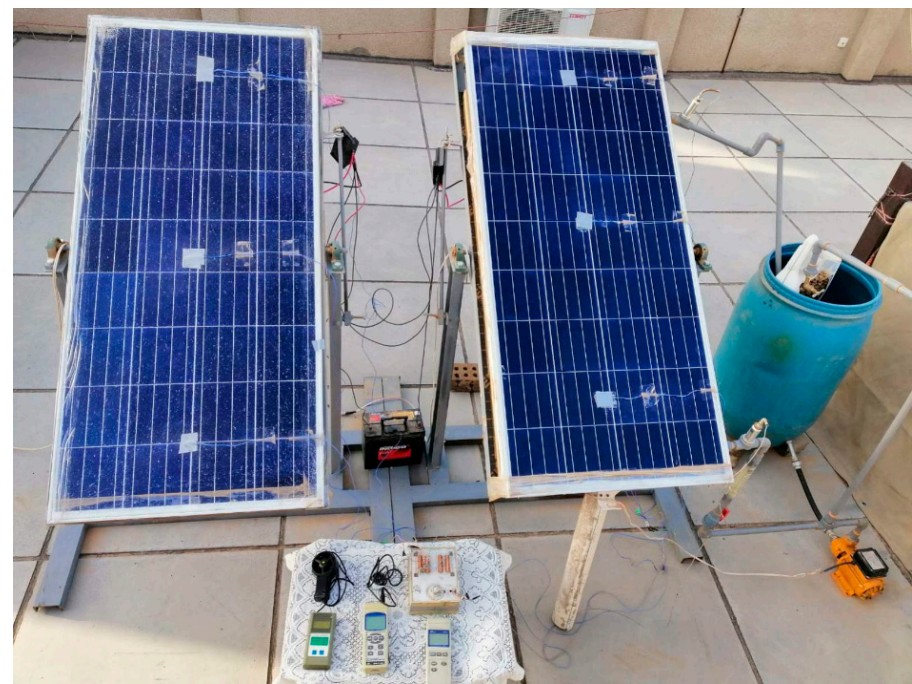

**Figure 2.** Experimental setup of the PV and PV/T system.

*2.2. Numerical Analysis*

Computational fluid dynamics (CFD) simulation techniques are efficient tool for representing a mechanical problem and analyzing their physical phenomena for engineering applications. In the present simulation, the commercial CFD software ANSYS 17.0 FLUENT software was used. This software is a general-purpose CFD software used to model fluid flow, heat and mass transfer, chemical reactions, and more. Fluent offers a modern, user-friendly interface that streamlines the CFD process from pre- to post-processing within a single window workflow. Geometry and modelling of the PV/T system was done in Solid works (ver. 2019) program. There are several steps that carried out in order to perform the simulations. The modelling is carried out in Solid work and, subsequently, the mesh is imported FLUENT k-$\varepsilon$ model for solving and post-processing as solar collector problem. This simulation is divided into two open-flow flat collector models according to the research objectives. A benchmark comprehensive numerical model to simulate the collector was developed to predict the thermal-hydrodynamic behavior of the system. The solution procedure of the model was achieved by computational coding in ANSYS environment. The models were validated by comparing the results with experimental measurements.

The CFD model is used to investigate heat loads for various shapes and parameters, such as inlet and outlet water temperatures and temperature distribution on the collector surface. Preparing a computational domain is the first step in CFD simulation. Model-C and model-S are three-dimensional open-flow flat collector models created in Solid work software. The geometrical models of open-flow flat collector are shown in Figure 3. These collectors are designed with a different bulges: cubic, and sphere. The dimension of collector is (670 $\times$ 1100 $\times$ 30 mm). The water inner and outer pipe diameters of the collector are 13 mm. Two shapes of aluminum bulges with 120 pieces are used. The first shape is cubic at dimensions 15 $\times$ 15 $\times$ 15 mm, while the second shape is a sphere at diameter 25 mm.

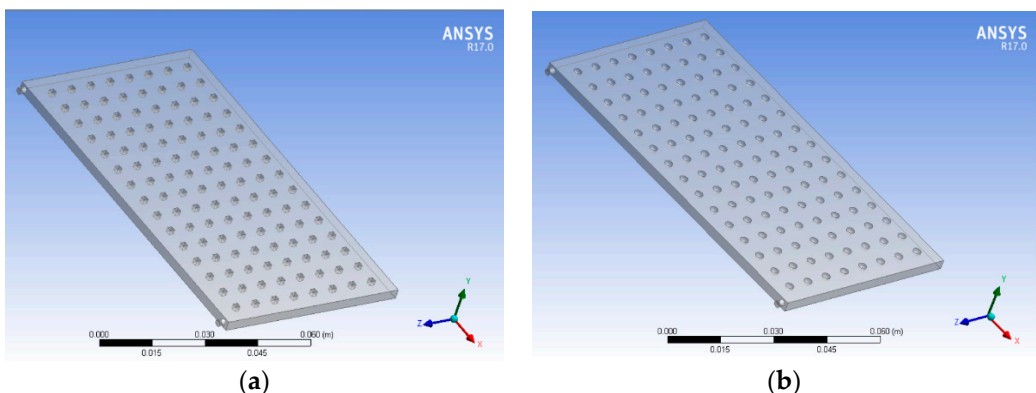

**Figure 3.** Computational domain prepared of open-flow flat collector; (**a**) model-C, and (**b**) model-S.

### 2.2.1. Computational Grid

The creation of the computational grid, which is made up of computational cells, is one of the most important steps in CFD. The governing equations are solved in computational cells. Unstructured meshing is used to mesh the collector domain with tetrahedral elements. Due to the complex geometry, unstructured tetrahedral meshing was chosen. A mesh independent solution is recommended for removing the influence of mesh size. When the grid size has no effect on the solution, the optimal grid size is chosen.

The 3D meshed geometrical flat collector model is shown in Figure 4. There are three mesh type to depict the collector can used: coarse, medium, and fine mesh. Figure 4 shows a closer look at the meshed bulges in the collector. The fine meshing scheme produced the most accurate and comparable results for heat transfer inside the collector, velocity, and temperature distribution behavior under influence of a bulge shape. As a result, the fine mesh was chosen for the simulations and result interpretation. Model-C had 96,914 nodes and 564,314 fine mesh elements, respectively. While fine mesh nodes and elements for Model-S were 94,844 and 523,711, respectively.

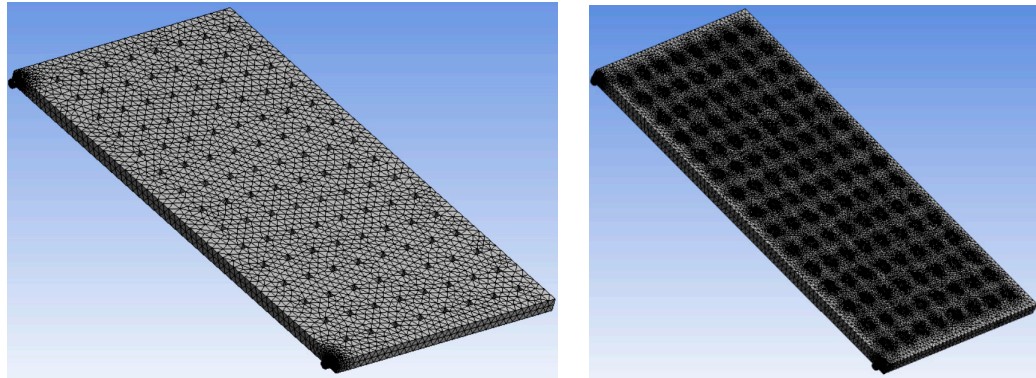

**Figure 4.** Open-flow flat collector geometrical mesh model, fine mesh, and a close-up of the bulge shape.

### 2.2.2. Solving Governing Equations

Under turbulent and steady conditions, the FPC model and domain were simulated in three dimensions. In 3D Cartesian coordinates, the governing equations of continuity, Naviers-Stokes, and thermal energy are as stated in equations [16,29,30]. The continuity equation is obtained by expressing each term in terms of velocity components:

Expressing each term in terms of velocity components gives the continuity equation:

$$\frac{\partial \rho}{\partial t} + \frac{\partial}{\partial x}(\rho u) + \frac{\partial}{\partial y}(\rho v) + \frac{\partial}{\partial z}(\rho w) = 0 \tag{1}$$

Momentum equation:

$$\rho \frac{D\vec{V}}{Dt} = \rho \vec{g} - \nabla \vec{p} + \frac{1}{3}\mu\nabla\left(\nabla.\vec{V}\right) + \mu\nabla^2\vec{V} \tag{2}$$

where $V$ is velocity vector, $g$ is gravity, and $\mu$ is viscosity.

Energy equation:

$$\rho c p \frac{DT}{Dt} = \nabla.k\nabla T + \beta T \frac{Dp}{Dt} + \mu\varnothing \tag{3}$$

where $k$ is thermal conductivity, $p$ is pressure, $\varnothing$ is a thermal energy source, and $\beta$ is the coefficient of thermal expansion, defined as:

$$\beta = -\frac{1}{\rho}\left[\frac{\partial p}{\partial T}\right]_p \tag{4}$$

Additionally, the dissipation function $\varnothing$ is associated with energy dissipation due to friction. It is portant in high-speed flow and for very viscous fluids [19]. In Cartesian coordinates $\varnothing$ is given by:

$$\varnothing = 2\left[\left(\frac{\partial u}{\partial x}\right)^2 + \left(\frac{\partial v}{\partial y}\right)^2 + \left(\frac{\partial w}{\partial z}\right)^2\right] + \left[\left(\frac{\partial u}{\partial y} + \frac{\partial v}{\partial x}\right)^2 + \left(\frac{\partial v}{\partial z} + \frac{\partial w}{\partial y}\right)^2 + \left(\frac{\partial w}{\partial x} + \frac{\partial u}{\partial z}\right)^2\right] - \frac{2}{3}\left(\frac{\partial u}{\partial x} + \frac{\partial v}{\partial y} + \frac{\partial w}{\partial z}\right)^2 \tag{5}$$

## 3. Results and Discussion

This section includes a discussion and analysis of our experimental and numerical results, which included two paths. The first path discusses the impact of the specifications of the new design on the performance of the solar collector, where the results of the model-S design, the model-C design, the change in collector surface temperatures, the amount of heat lost, and the thermal collector efficiency will be presented, and these specifications are compared for both designs. The second path discusses numerical results, such as the effect of water flow rate on the surface temperature of the collector, and its impact on the performance of the solar collector, and the comparison of experimental and theoretical results.

*3.1. Experimental Results*

3.1.1. Effect of Collector Design on PV/T System Performance

The temperature of the cell surface changes when different designs are used, as shown in Figure 5. The uncooled cell temperature is known to be too high due to a lack of cooling, resulting in a decrease in electrical efficiency. It conducted some practical experiments with two designs, bulge S and bulge C. At 1:00 p.m., the uncooled cell temperature was 71 °C. Their effects on the system as the cell temperature decreased using water cooling and a flow rate of 1.5 L per minute were observed. According to the results, there was a slight improvement in swell C due to cooling, and the use of swell S. The ratio of uncooled cell to bulge C and bulge S was 8.4% and 9.8%, respectively.

Figure 6 shows the cooling with water and using different shapes for the purpose in order to reduce cell temperatures and raise electrical efficiency, it receives thermal results, such as hot water can be used in some applications as the home requirements. It notices some changes to the system using two different designs, such as Bulge S and Bulge C where the uncooled cell temperature is very high due to the lack of cooling 71 °C at 1:00 p.m. While the cooling at the flow rate (1.5 L/min), it notices that there is a clear improvement as the collector temperature decreased, according to Figure 6, in the bulge S, while the improvement in the bulge C is less, and the reason is that the bulge S has a streamlined design and the fluid is in contact with it from all sides, which increases the suction process. Where the percentage of the difference in temperature between uncooled cell and the model-C and the model-S is at 1:00, it was 23.9% and 25.3%, respectively. From the results,

it notes that the higher the radiation, the higher the heat output and thus the higher the heat gain.

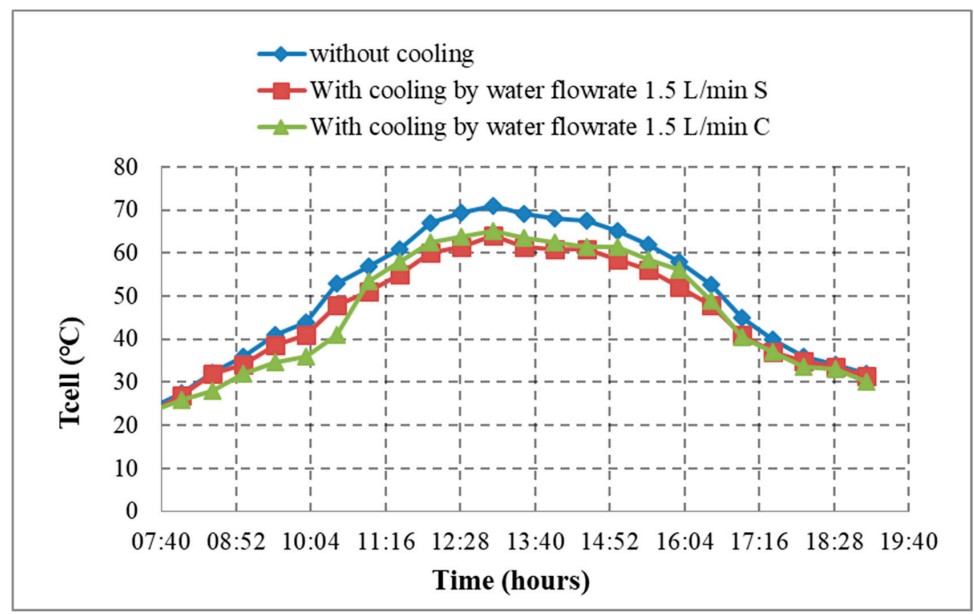

**Figure 5.** Variation of cell surface temperature with time during the day at the flow rate 1.5 L/min for bulge S and bulge C.

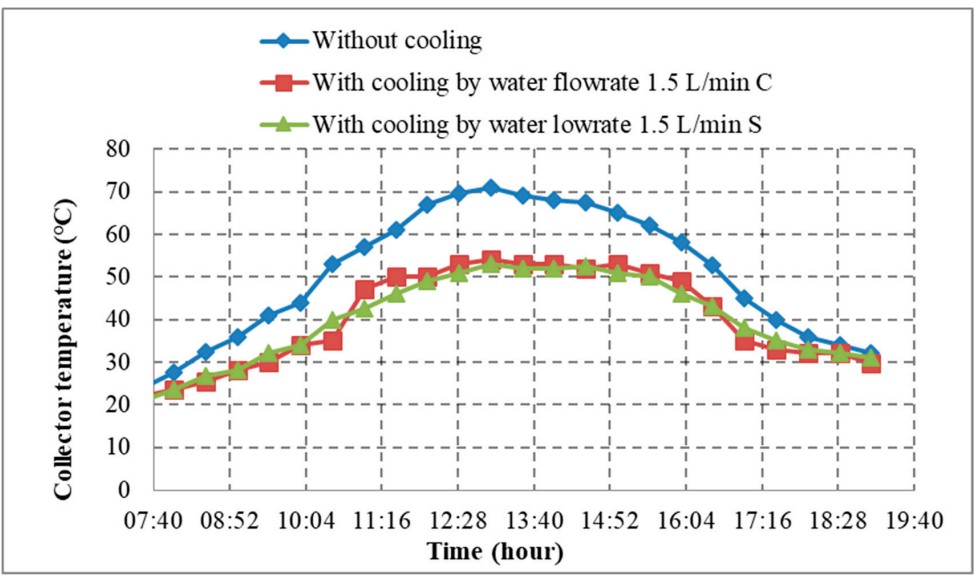

**Figure 6.** Variation of surface collector temperature with time during the day at the flow rate 1.5 L/min for bulge S and bulge C.

Figure 7 represents the experiments and the beginning with daily measurements during the day with entering some parameters into the system and using two different designs, such as the bulge S and the bulge C at a certain flow rate, the heat gain is determined first by the input and output of the water, and, in general, the amount of radiation entering the photovoltaic cell performs the most important role. Part of this radiation can be dissipated, and the other part can be reflected, but the largest part enters the cell, then turns into heat, meaning 75% of the heat enters the collector. Thus, when comparing the SS model with the model-C for the same flow rate when using a flow rate of 3.5 L per minute, it notices that the temperature difference for the bulge S is better than the bulge C, and the reason for this is that the bulge S has a spherical design, which gives more suction capacity where the percentage

of the difference in temperature for the model-C and the model-S is at 12:30, it is the ratio (1.01%). We note that the higher the rate of flow, the difference in temperature decreases.

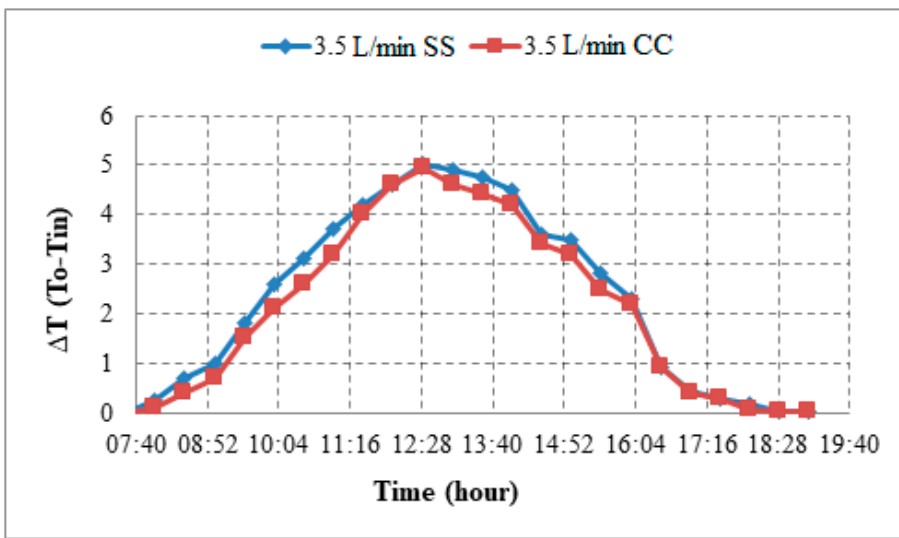

**Figure 7.** Variation of difference water temperature with time during the day at the flow rate 3.5 L/min for bulge S and bulge C.

Figure 8 when entering parameters into the system, such as the change in the rate of flow and the use of some different designs, when entering and leaving the water, it found there are heat gains, and these heat gains depend mainly on the difference in temperature. It noted that with more radiation, the direct ($\eta$th) increases, when cooling at a flow rate of 3.5 L/min, and the use of the two different designs, such as the bulge S and the bulge C. It noticed that there is an improvement in the bulge S that is better than the bulge C. The reason for this is that, when using the bulge S design, the thermal suction is higher, which gives a higher thermal efficiency, and, from the experiments results, the percentage difference between the model-C and the model-S is at 1:00, and the ratio is 6.06%. It is known that the higher the flow rate, the higher the thermal efficiency, while the higher the flow rate, the lower the temperature difference, according to the amount of radiation.

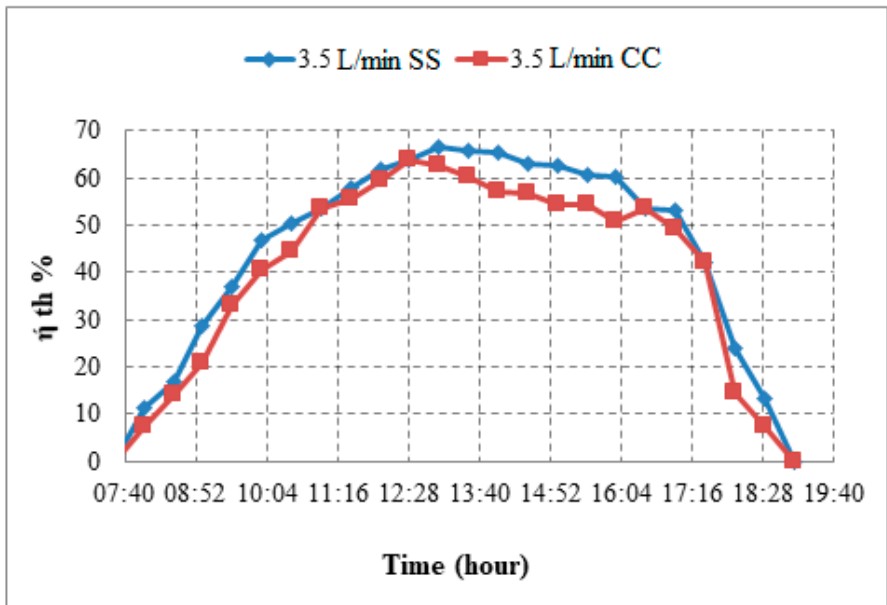

**Figure 8.** Variation of thermal efficiency with time during the day at the flow rate 3.5 L/min for bulge S and bulge C.

3.1.2. Effect of Water Flow Rate on PV/T System Performance

Figure 9 shows the thermal efficiency of PV/T can be known by the input of solar energy, and it can be converted into thermal gain. Upon investigation, it found noticeable changes in the thermal efficiency ($\acute{\eta}$th) of the solar collector (PVT) with radiation (G) at different flow rates. The radiation is directly proportional to the ($\eta$th). When the flow rate is increased, the thermal efficiency increases, and thus, at the highest radiation, the highest thermal efficiency is achieved as the thermal efficiency increases at the following flow rates 2 L/min, 2.5 L/min, and 3.5 L/min, respectively. The increases were 11.9%, 25.5%, and 35.6%, respectively, in relation to the flow rate 1.5 L/min at 12:30 p.m. The thermal efficiency can be extracted from the results discussed on the PV/T system. Both $\eta$el and $\acute{\eta}$th increased with the increase in flow rate. The current work is similar to that of Abdullah et al. [18], where it is thermal efficiency ($\acute{\eta}$th) from 42.46% to 45.60% as in AL-Kayiem et al. [21].

Figure 10 shows the thermal efficiency of PV/T can be known by the input of solar energy and it can be converted into thermal gain. Upon investigation, it found noticeable changes in the thermal efficiency ($\acute{\eta}$th) of the solar collector (PVT) with radiation (G) at different flow rates. When the flow rate is increased, the thermal efficiency increases, and thus, at the highest radiation, the highest thermal efficiency is achieved as the thermal efficiency increases at the following flow rates 2 L/min, 2.5 L/min, and 3.5 L/min, respectively. The increases were 7.8%, 16.6%, and 27.3%, respectively, in relation to the flow rate 1.5 L/min at 1:00 p.m. The thermal efficiency can be extracted from the results discussed on the PV/T system. Both $\eta$el and $\acute{\eta}$th increased with the increase in flow rate. The current work is similar to that of Abdullah et al. [18], where the thermal efficiency ($\acute{\eta}$th) is from 42.46% to 45.60%, as in AL-Kayiem et al. [21].

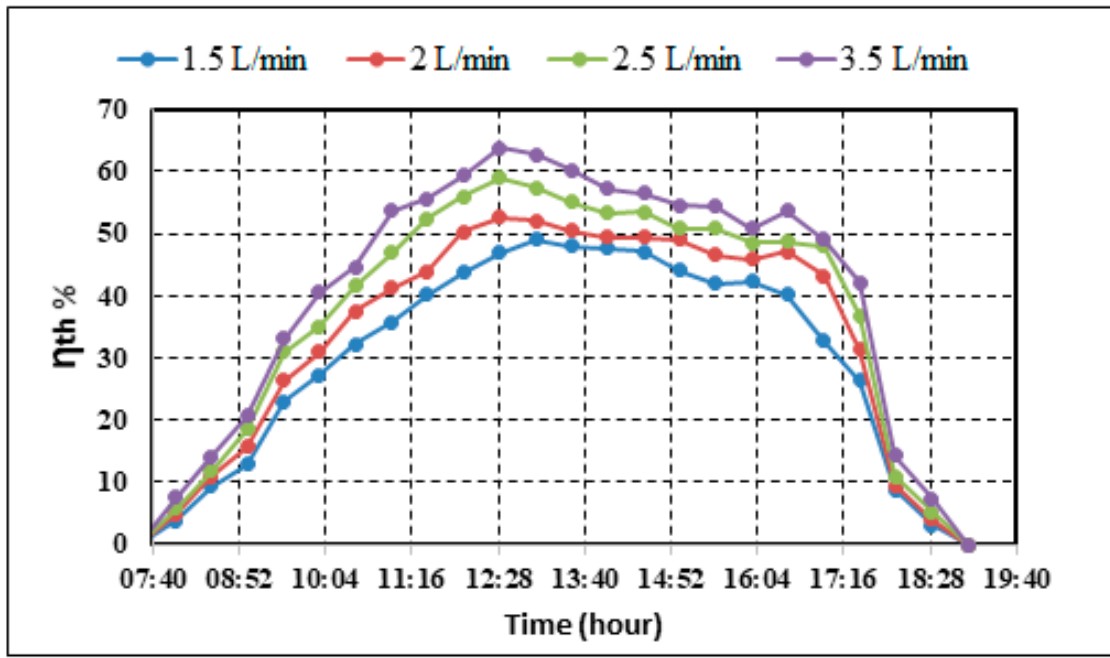

**Figure 9.** Variation of thermal efficiency with time during the day at different the flow rates for bulge C.

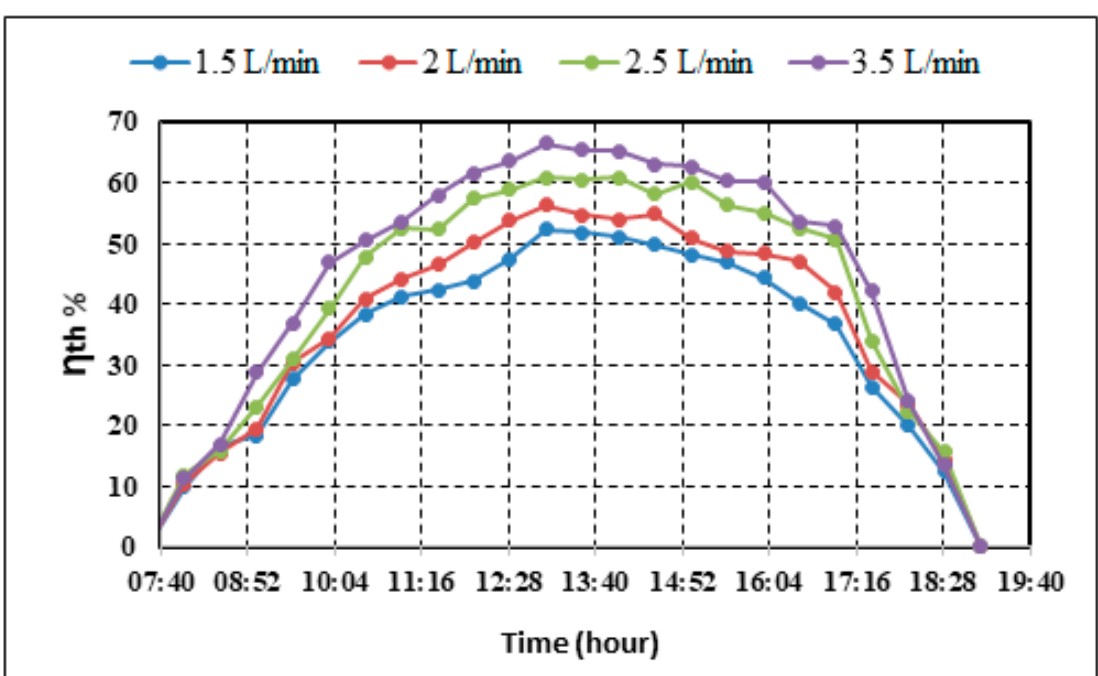

**Figure 10.** Variation of thermal efficiency with time during the day at different the flow rates for bulge S.

### 3.2. Numerical Results

Figure 11 shows the contours of temperature distribution the collector for flow rate 1.5 L/min for model-C (a) at 8 a.m., (b) at 10 a.m., (c) at 1 p.m., and (d) at 6 p.m. It notes from the results that when cooling with water and entering the fluid into the collector, there was a decrease in the entry temperatures, but at the exit, the temperature of the fluid increased due to the suction process, and the fluid absorbing heat when using the bulge C design from the figure. It noticed the rise in the temperature of the contour as it went to the top.

Figure 12 represents the relationship between (ΔT) with time during the day where it notices from the numerical program when cooling with water and releasing different flow rates that the higher the flow rate, the lower the temperature difference, while the highest difference is at the lowest flow rate. Numerical results showed that the difference in inlet and outlet temperatures decreases with increasing water flow rates, with the decreasing percentages being 11.8%, 12.7%, and 12.9%, respectively, when changing the flow rates to 2 L/min, 2.5 L/min, and 3.5 L/min, respectively.

Figure 13 temperature distribution contours in the collector for a flow rate of 1.5 L/min for model-S (a) at 8 a.m., (b) at 10 a.m., (c) at 1 p.m., and (d) at 6 p.m. From the theoretical program, we note that when cooling with water and entering the fluid into the collector, there was a decrease in the entry temperatures, but at the exit, the temperature of the fluid increased due to the suction process and the fluid absorbing heat when using the bulge S design. From the figure, we noticed the rise in the temperature of the contour as we went to the top.

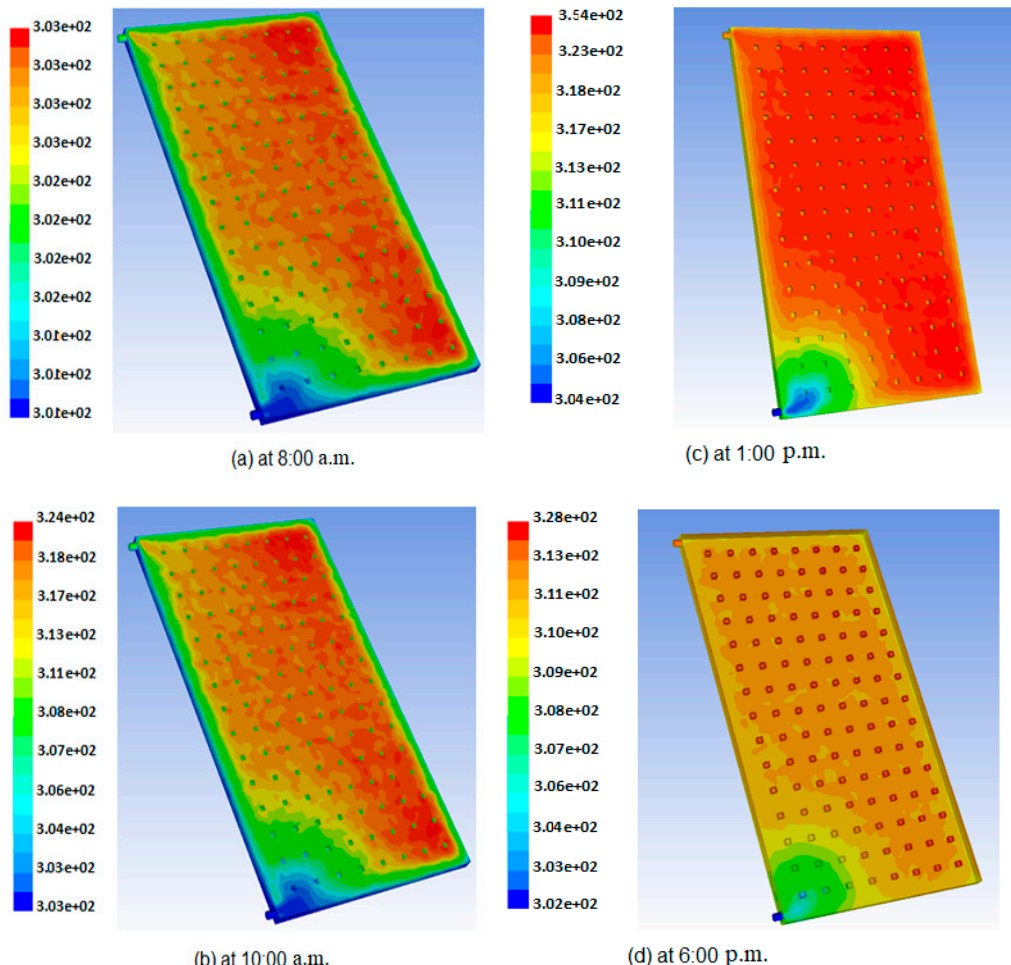

**Figure 11.** Contours of temperature distribution for Model-C collector (1.5 L/min flow rate); (**a**) at 8 a.m., (**b**) at 10 a.m., (**c**) at 1 p.m., and (**d**) at 6 p.m.

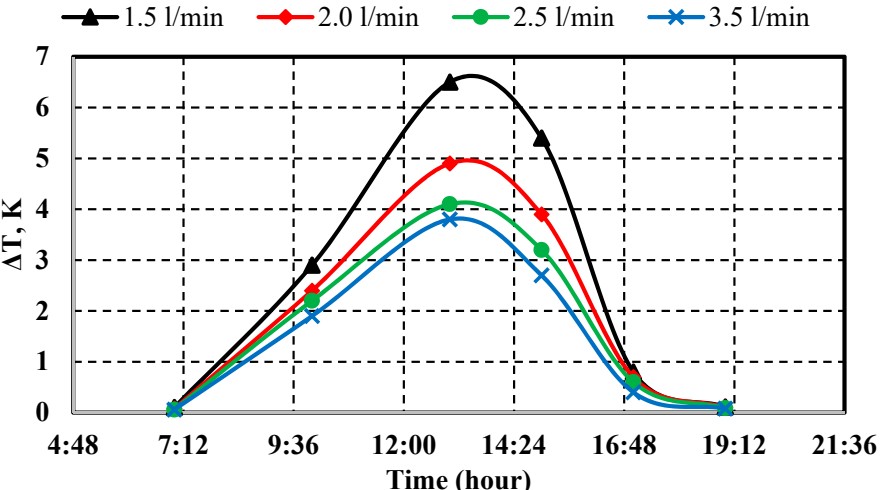

**Figure 12.** Simulated difference water temperature between the inlet and output for Model-C collector at different flow rates.

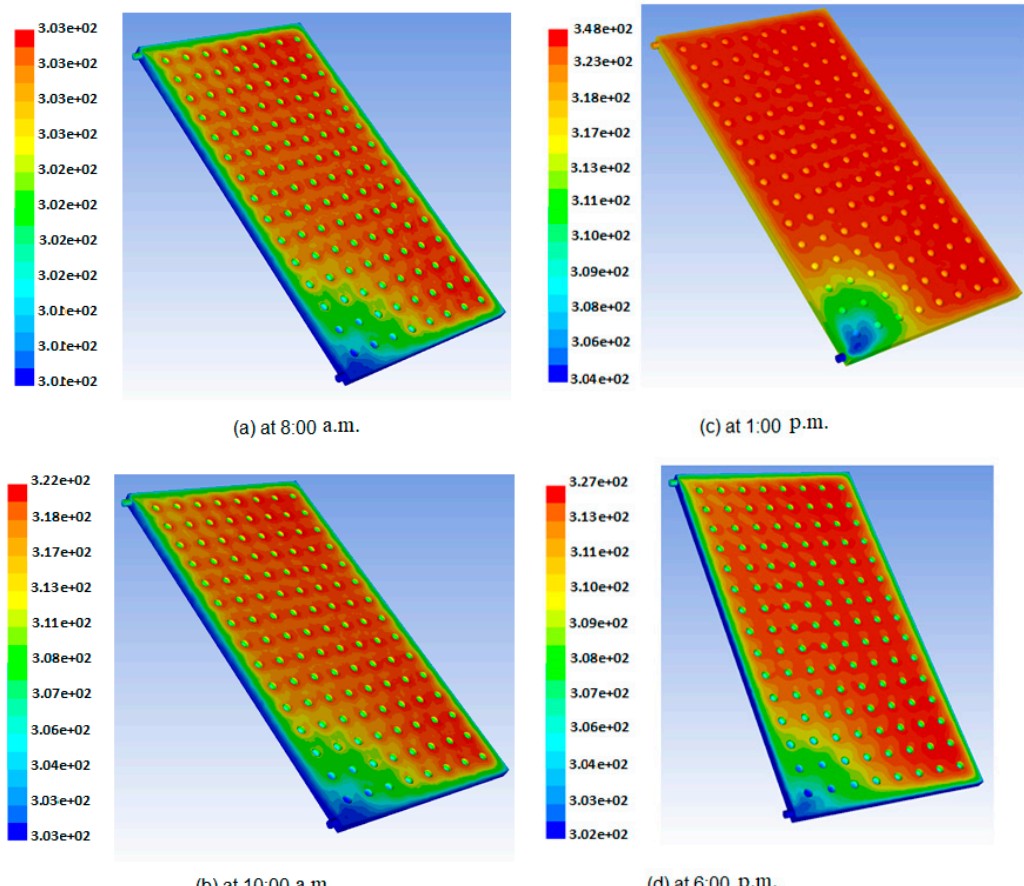

**Figure 13.** Contours of temperature distribution for Model-S collector (1.5 L/min flow rate); (**a**) at 8 a.m., (**b**) at 10 a.m., (**c**) at 1 p.m., (**d**) at 6 p.m.

Figure 14 represents the relationship between ($\Delta$T) with time during the day. We notice from the theoretical program that when cooling with water and releasing different flow rates, that the higher the flow rate, the lower the temperature difference, while the highest difference is at the lowest flow rate. Theoretical results show that the difference in inlet and outlet temperatures decreases with increasing water flow rates, with the decreasing percentages being 9.8%, 10.7%, and 11.2%, respectively, when changing the flow rates to 2 L/min, 2.5 L/min, and 3.5 L/min, respectively.

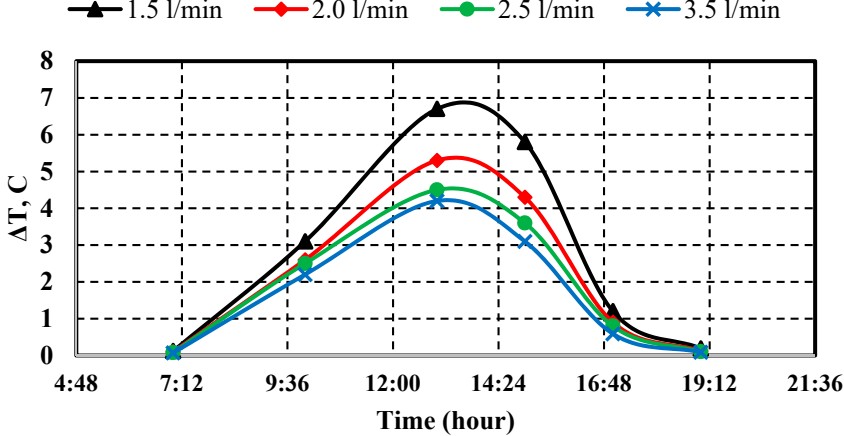

**Figure 14.** Simulated difference water temperature between the inlet and output for Model-S collector at different flow rates.

### *3.3. Validation of Numerical Result of Different Water Temperature Model-C and Model-S*

Figure 15 represents the relationship between ($\Delta$T) with time during the day and a comparison between the theoretical and practical for the flow rates of (1.5 L/min and 3.5 L/min), where it was noticed that when cooling with water and releasing different flow rates, that the higher the flow rate, the lower the temperature difference, while the difference was highest at the lowest flow rate. The theoretical and practical results showed that the difference in the entry and exit temperatures decreases with the increase in water flow rates. The total difference (error) percentage between the numerical and experimental was 13.8% for 1.5 L/min and 9.8% for 3.5 L/min for model-C, and the total difference (error) percentage between the numerical and experimental was 11.6% for 1.5 L/min and 8.7% for 3.5 L/min for model-S.

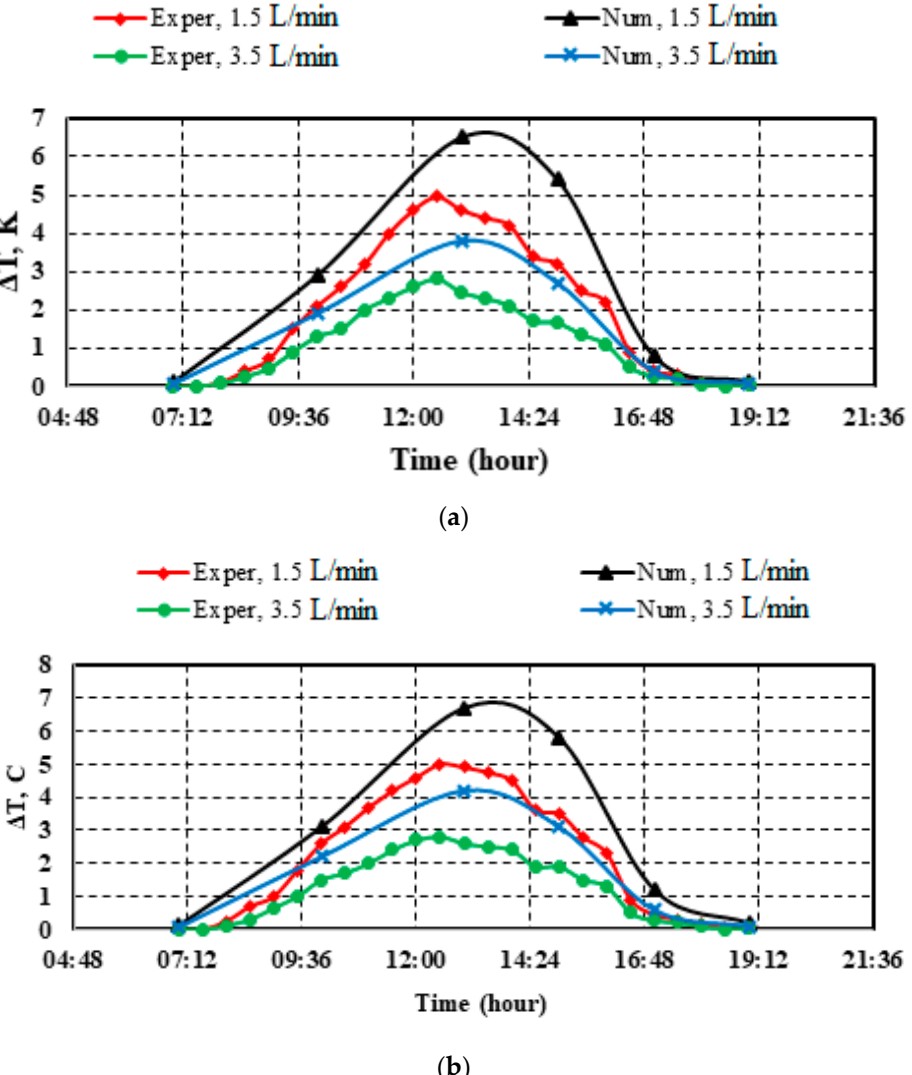

(a)

(b)

**Figure 15.** Different water temperatures between the inlet and output at 1.5 and 3.5 L/min flow rates; (**a**) for Model-C, (**b**) for Model-S.

## 4. Conclusions

This work describes an experimental and numerical analysis to study the influence of water flow rates through an open flow cooling collector with a new two models designed for PV/T cooling. The temperature distribution for different water flow rates with 120 sphere/or cubic bulges fixed on the bottom of the upper cover of the cooling collector have been studied. The following conclusions are drawn:

- ■ Results appeared with the effect of collector design (bulge shape) on the PV/T system performance and PV panel temperature. It was the percentage of difference temperature with the uncooled PV panel 8.4% and 9.8% for model-C and Model-S, at 1:00 p.m., while the performance of the PV panel increased to 23.9% and 25.3% for both models, respectively, at 1:00 p.m.
- ■ Effect of collector design (bulge shape) on PV/T system performance relation to ($\Delta$T), where the percentage of the difference in model-C and model-S at 12:30 was 1.01%. The effect of collector design (bulge shape) on PV/T system performance on heat gained, where the percentage of the different for the model-C and the model-S at 1:00 was 6.08%.
- ■ Effect of collector design (bulge shape) on PV/T system performance relation to thermal efficiency, where the percentage of the difference between for the model-C and the model-S at 1:00 is the ratio of 6.06%.
- ■ The numerical results showed that the difference in inlet and outlet temperatures were decreased with increasing water flow rates by 7.8%, 11.7%, and 14.9%, respectively, when changing the flow rates to 2.0 L/min, 2.5 L/min, and 3.5 L/min, respectively.
- ■ The average temperature of the upper surface of the cooling collector decreases with the increase in water rates by 3.2%, 4.8%, and 5.9%, respectively, when changing the flow rates to 2 L/min, 2.5 L/min, and 3.5 L/min, respectively.

Through this work recommend to improvement the PV/T system performance. Further investigations using different types and completely different geometry—for example, a long flat rib, flow turbulator, etc.

## 5. Directions for Further Research

Generally, high operating temperatures cause the shorten life cycle of the PV module due to damaging the module material. This situation has been substantially eliminated by cooling the PV cell. For this reason, prolonged payback time of the PV system and shortening the life of the materials used in PV modules are among the causes of occurrence high temperature. As a result of the foregoing, the research takes two paths to develop the PV/T system:

1. Developing a PV/T system by searching for the optimal design of the collector with high thermal efficiency by using methods that increase heat transfer, such as fins or the use of porous materials.
2. Developing a PV/T system by improving the enhancement thermal fluid properties by using Nano additives.

**Author Contributions:** M.G.A. and B.A.S., contributed to the experimental part including the fabrication and setup of the PV/T system and data collection, and the theoretical part represented by numerical modeling. While E.G. and H.A.A.W., helped with numerical analysis, results analysis, and article writing. All authors have read and agreed to the published version of the manuscript.

**Funding:** This research received no external funding.

**Institutional Review Board Statement:** Not applicable.

**Informed Consent Statement:** Informed consent was obtained from all subjects involved in the study.

**Data Availability Statement:** Not applicable.

**Conflicts of Interest:** The authors declare no conflict of interest.

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
