# Peer review of "Performance Analysis of an Open-Flow Photovoltaic/Thermal (PV/T) Solar Collector with Using a Different Fins Shapes"

_sustainability, doi:10.3390/su15053877_

Round 1

Reviewer 1 Report

This paper entitled “Prediction of Open-Flow Flat Collector Performance for PV Panel Thermal Control Application”; can be considered for publication after minor revisions explained below.

1)      Needs editing for better understanding.

2)      In introduction section, In the introduction, you should refer to the article of

1- Dembeck-Kerekes, T, Fine, JP, Friedman, J, Dworkin, SB & McArthur, JJ 2019, 'Performance of variable flow rates for photovoltaic-thermal collectors and the determination of optimal flow rates', Solar Energy, vol. 182, pp. 148-160.

2- International Journal of Physics. 2018, 6(3), 93-98

3- Hussain, M. Imtiaz, and Jun-Tae Kim. 2020. "Performance Evaluation of Photovoltaic/Thermal (PV/T) System Using Different Design Configurations" Sustainability 12, no. 22: 9520. https://doi.org/10.3390/su12229520

4- International Journal of Partial Differential Equations and Applications, 2017, Vol. 5, No. 1, 19-25

3)      To attract the reader, it is better to choose another title

4)      How is the Reynolds number in the channel calculated? How is the velocity or mass flow rate measured?

5)         The abstract needs to be revised to point out the importance of this study. 

Author Response

Respond to Reviewer 1 comments

This paper entitled “Prediction of Open-Flow Flat Collector Performance for PV Panel Thermal Control Application”; can be considered for publication after minor revisions explained below.

  • Needs editing for better understanding.

Authors: The reviewer comment is valid, and it has been considered.

  • In introduction section, you should refer to the article of

1- Dembeck-Kerekes, T, Fine, JP, Friedman, J, Dworkin, SB & McArthur, JJ 2019, 'Performance of variable flow rates for photovoltaic-thermal collectors and the determination of optimal flow rates', Solar Energy, vol. 182, pp. 148-160.

2- International Journal of Physics. 2018, 6(3), 93-98

3- Hussain, M. Imtiaz, and Jun-Tae Kim. 2020. "Performance Evaluation of Photovoltaic/Thermal (PV/T) System Using Different Design Configurations" Sustainability 12, no. 22: 9520. https://doi.org/10.3390/su12229520

4- International Journal of Partial Differential Equations and Applications, 2017, Vol. 5, No. 1, 19-25

Authors: Done. The references have been added to introduction.

  1. Dembeck-Kerekes, T., et al., Performance of variable flow rates for photovoltaic-thermal collectors and the determination of optimal flow rates. Solar Energy, 2019. 182: p. 148-160.
  2. Hussain, M.I. and J.-T. Kim, Performance evaluation of photovoltaic/thermal (PV/T) system using different design configurations. Sustainability, 2020. 12(22): p. 9520.
  3. Heidarian, A., H. Ghassemi, and P. Liu, Drag reduction by using the microriblet of sawtooth and scalloped types. International Journal of Physics, 2018.
  4. Mahmoodi, K., H. Ghassemi, and A. Heydarian, Solving the Nonlinear Two-Dimension Wave Equation Using Dual Reciprocity Boundary Element Method. International Journal, 2017. 5(1): p. 19-25.
  • To attract the reader, it is better to choose another title

Authors: the reviewer comment is valid, and it has been considered. We changed title to

“Performance Analysis of an Open-Flow Photovoltaic/Thermal (PV/T) Solar Collector with Using a Different Fins Shapes”

  • How is the Reynolds number in the channel calculated? How is the velocity or mass flow rate measured?

Authors: the reviewer comment is good, we measured water volumetric flow rates. (l/min)

About calculate Reynolds number “ In channel flow, the Reynolds number (Re) is defined as Re = Uh/νk, where U is the centreline velocity, h is the half-height of the channel, and νk is the kinematic viscosity of the fluid.”

  • The abstract needs to be revised to point out the importance of this study.

Authors: the reviewer comment is valid, and it has been considered.

“Generally, photovoltaic panels convert solar energy into electricity using semiconductor materials in their manufacture, by converting energy into electricity by absorbing heat from solar radiation, which requires reducing the heat of these panels to improve the efficiency of electricity generation. Therefore, the issue of cooling photovoltaic panels became one of the objectives that were addressed in many studies, while cost reduction was the most important concern in the manufacture of these panels, followed by low energy consumption. In this work, the performance analysis for PV panels was achieved through using two models (Model-C and Model-S) of open-flow flat collector improves the cooling process for PV panel. The investigations of open-flow flat collector have been performed and analyzed using experimental and numerical methods. The simulation analysis was carried out by ANSYS FLUENT 17.0 software with two open-flow flat collector modules. Results appeared the effect of collector design (fin shape) on PV/T system performance and PV panel temperature, it was the percentage of difference temperature with uncooled PV panel 8.4% and 9.8% for Model-C and Model-S, at 1:00 pm. While, the performance of PV panel increased to 23.9% and 25.3% with both models respectively at (1:00 pm). The evaluation result demonstrates that the performance of PV/T system increased, also the fins in open-flow collector helped to system enhances.”

Reviewer 2 Report

TITLE

The title of the manuscript is appropriate to the research described in the manuscript.

 Abstract

The abstract of this manuscript is clear and concrete on the developed topic. The information contained in this section is correctly structured and supported by the information developed in the research that supports this manuscript. The information presented in this section could be improved if some data on the prediction of open-flow flat collector performance for PV panel thermal control application in other investigations.

 Introduction

The introduction is clear and adequately related to the topic that is being developed in the manuscript. But it is important to improve the number of cited references and their timeliness. It is important to improve the scientific support of the contributions presented in this manuscript.

 Materials and Methods

This section is presented in a clear manner and specifi the materials and methods used in the research describe in the manuscrit. And the parameters of the on the prediction of open-flow flat collector performance for PV panel thermal control application are describe with clarity and technical fundament.

 Results

This section is not defined as marked by the Journal Guidelines, so it is suggested that they adhere to the format.

The results presented in this section are shown in a clear and orderly manner. In addition, they support the conclusions presented in the manuscript.

The content of this section can be improved if data from other similar investigations are included to allow comparison of the prediction of open-flow flat collector performance for PV panel thermal control application. This will more adequately show the novelty of the results presented in this manuscript.

 Discussion

This section is not individually defined in the writing as marked by the Journal's guidlines. Authors are advised to adhere to the Journal format.

 Conclusion

The information presented in the conclusions of the manuscript is clear and concise. It adequately describes the experimental information developed in the research described in the manuscript. The conclusions have a solid basis regarding the premise of the manuscript and the experimental information presented.

 References

The number of references cited in this manuscript is low. This prevents comparing the timeliness of the information described in this writing. The distribution of references is 69% older than 5 years, 15% between 5-10 years and the remaining 12% older than 10 years.

It is recommended that the authors increase the number of references cited in this manuscript to show the novelty of the research embodied here. In addition, they must maintain the distribution over time of the references they have shown up to now.

The premise of improving the number of cited references in this manuscript is intended to show the relevance of this research focused on the effect of Open-Flow Flat Collector Performance.

It is important to review the format of the references from number 23 to number 26 and, if necessary, correct them.

Author Response

Respond to Reviewer 2 comments

TITLE

The title of the manuscript is appropriate to the research described in the manuscript.

Authors: the reviewer comment is valid, and it has been considered. We changed title to

“Performance Analysis of an Open-Flow Photovoltaic/Thermal (PV/T) Solar Collector with Using a Different Fins Shapes”

 Abstract

The abstract of this manuscript is clear and concrete on the developed topic. The information contained in this section is correctly structured and supported by the information developed in the research that supports this manuscript. The information presented in this section could be improved if some data on the prediction of open-flow flat collector performance for PV panel thermal control application in other investigations.

Authors: the reviewer comment is valid, and it has been considered.

“Generally, photovoltaic panels convert solar energy into electricity using semiconductor materials in their manufacture, by converting energy into electricity by absorbing heat from solar radiation, which requires reducing the heat of these panels to improve the efficiency of electricity generation. Therefore, the issue of cooling photovoltaic panels became one of the objectives that were addressed in many studies, while cost reduction was the most important concern in the manufacture of these panels, followed by low energy consumption. In this work, the performance analysis for PV panels was achieved through using two models (Model-C and Model-S) of open-flow flat collector improves the cooling process for PV panel. The investigations of open-flow flat collector have been performed and analyzed using experimental and numerical methods. The simulation analysis was carried out by ANSYS FLUENT 17.0 software with two open-flow flat collector modules. Results appeared the effect of collector design (fin shape) on PV/T system performance and PV panel temperature, it was the percentage of difference temperature with uncooled PV panel 8.4% and 9.8% for Model-C and Model-S, at 1:00 pm. While, the performance of PV panel increased to 23.9% and 25.3% with both models respectively at (1:00 pm). The evaluation result demonstrates that the performance of PV/T system increased, also the fins in open-flow collector helped to system enhances.”

 Introduction

The introduction is clear and adequately related to the topic that is being developed in the manuscript. But it is important to improve the number of cited references and their timeliness. It is important to improve the scientific support of the contributions presented in this manuscript.

Authors: the reviewer comment is valid, and it has been considered. We added 4 references.

Materials and Methods

This section is presented in a clear manner and specify the materials and methods used in the research describe in the manuscript. And the parameters of the on the prediction of open-flow flat collector performance for PV panel thermal control application are describe with clarity and technical fundament.

Authors: Done.

 Results

This section is not defined as marked by the Journal Guidelines, so it is suggested that they adhere to the format.

The results presented in this section are shown in a clear and orderly manner. In addition, they support the conclusions presented in the manuscript.

The content of this section can be improved if data from other similar investigations are included to allow comparison of the prediction of open-flow flat collector performance for PV panel thermal control application. This will more adequately show the novelty of the results presented in this manuscript.

 Authors: the reviewer comment is valid, and it has been considered.

Discussion

This section is not individually defined in the writing as marked by the Journal's guidlines. Authors are advised to adhere to the Journal format.

Authors: the reviewer comment is valid.

Conclusion

The information presented in the conclusions of the manuscript is clear and concise. It adequately describes the experimental information developed in the research described in the manuscript. The conclusions have a solid basis regarding the premise of the manuscript and the experimental information presented.

Authors: Done.

References

The number of references cited in this manuscript is low. This prevents comparing the timeliness of the information described in this writing. The distribution of references is 69% older than 5 years, 15% between 5-10 years and the remaining 12% older than 10 years.

It is recommended that the authors increase the number of references cited in this manuscript to show the novelty of the research embodied here. In addition, they must maintain the distribution over time of the references they have shown up to now.

The premise of improving the number of cited references in this manuscript is intended to show the relevance of this research focused on the effect of Open-Flow Flat Collector Performance.

It is important to review the format of the references from number 23 to number 26 and, if necessary, correct them.

Authors: Done. The references have been added to introduction.

  1. Dembeck-Kerekes, T., et al., Performance of variable flow rates for photovoltaic-thermal collectors and the determination of optimal flow rates. Solar Energy, 2019. 182: p. 148-160.
  2. Hussain, M.I. and J.-T. Kim, Performance evaluation of photovoltaic/thermal (PV/T) system using different design configurations. Sustainability, 2020. 12(22): p. 9520.
  3. Heidarian, A., H. Ghassemi, and P. Liu, Drag reduction by using the microriblet of sawtooth and scalloped types. International Journal of Physics, 2018.
  4. Mahmoodi, K., H. Ghassemi, and A. Heydarian, Solving the Nonlinear Two-Dimension Wave Equation Using Dual Reciprocity Boundary Element Method. International Journal, 2017. 5(1): p. 19-25.

Also, we re-arrangement references from 23 to 26. And became 25 to 28.

Reviewer 3 Report

The presented article is a numerical and experimental study of solar photovoltaic thermal modules of a planar design with various geometric protrusions on the module heatsink. The topic of the article is relevant and may be of interest to specialists and researchers in the fields of solar energy and modeling. The authors have carried out theoretical and experimental studies, obtained interesting results, however, as comments and recommendations, several points should be noted:

1. There is an extra comma on line 25. On line 34, the abbreviation is duplicated. Simultaneous citation of several sources (for example, [1-6], [16-20], [21-26]) should be avoided and refer to each source separately and specifically. The authors should decipher the indicators of the graphs in Figures 5 and 6 (Tpv, Qw). Also, the authors should describe in more detail the differences between Figures 5 and 6. In Figure 9, all coolant flow rates for the corresponding graphs should be indicated. All parameters used in equations 3.1 - 3.5 should be described and characterized following each formula. On line 252, the authors should clarify "it get thermal results such as hot water and some other uses" - what other uses?

2. The direction considered by the authors is actively developing and a large number of different solar photovoltaic thermal modules have been developed - thus, the authors should increase the literature review by considering modern and relevant articles from high-ranking world publications, after which the list of sources used will also increase, which will also strengthen the article.

3. Authors should more clearly indicate the scientific novelty of the structures and methods used.

4. The authors should justify the choice of software used for research in the work.

5. What justified the choice of geometry and the distribution of ledges? Could their other forms and distributions be more effective? It would be advisable to carry out the experiment also with a completely different geometry - for example, a long flat rib, flow turbulators, etc., since the differences between the considered configurations are not so significant.

6. How and with the help of what did the ribs attach to the metal sheet - what is the thickness and thermal conductivity of the connecting layer? Is a metal sheet added to a standard PV module? What is the thermal conductivity between these layers - after all, standard photovoltaic modules are not designed for hybrid operation. The polysloxane compound used to seal photovoltaic converters serves to increase the optical efficiency and better electrical efficiency of photovoltaic converters when they work with coolants (for example, DOI: 10.4018 / IJEOE. 2020040106) - it is advisable for the authors to keep in mind this manufacturing technology, which is relevant in photovoltaic thermal modules.

7. Authors should add drawings or sketches of the used module designs with the principle of their operation.

8. Judging by Figure 7, the maximum heating of the coolant is only 5 degrees? Then it makes sense to significantly improve the design of the module and its thermal insulation.

9. What is the optimal coolant flow rate for the developed modules? What is the overall efficiency of the developed modules? The developed modules have an electrical efficiency 23.9% and 25.3% (line 402)?

10. Authors should indicate how and by what formulas the thermal efficiency of modules was calculated.

11. How were the results obtained in Figures 12 and 14? Areas of overheating and stagnation are visible, which indicates the need for further optimization of the module heatsink designs (for example, DOI: 10.4018/978-1-5225-3867-7.ch004), which should be carried out in further research by the authors.

12. Authors should pay attention to writing English in the work.

13. The authors should add a section "Directions for further research", where they indicate the planned further work in this direction. For the modules under consideration, it is advisable to create special solar photovoltaic thermal modules with a pre-developed manufacturing technology and special sealing components of photovoltaic converters, as well as more versatile optimization of the design of the photovoltaic thermal module radiator.

14. Also, the authors should add a paragraph at the end of the article indicating the contribution of each author to the content of the work.

In general, the presented article leaves a positive impression, however, it is not without flaws. After eliminating these remarks and taking into account the recommendations made, the presented article may be of interest to readers of the journal "Sustainability".

Author Response

Respond to Reviewer 3 comments

  1. There is an extra comma on line 25. On line 34, the abbreviation is duplicated. Simultaneous citation of several sources (for example, [1-6], [16-20], [21-26]) should be avoided and refer to each source separately and specifically. The authors should decipher the indicators of the graphs in Figures 5 and 6 (Tpv, Qw). Also, the authors should describe in more detail the differences between Figures 5 and 6. In Figure 9, all coolant flow rates for the corresponding graphs should be indicated. All parameters used in equations 3.1 - 3.5 should be described and characterized following each formula. On line 252, the authors should clarify "it get thermal results such as hot water and some other uses" - what other uses?

Authors: all reviewer comments are valid, and it have been considered. We do all changes.

  1. The direction considered by the authors is actively developing and a large number of different solar photovoltaic thermal modules have been developed - thus, the authors should increase the literature review by considering modern and relevant articles from high-ranking world publications, after which the list of sources used will also increase, which will also strengthen the article.

Authors: Done. The references have been added to introduction.

  1. Dembeck-Kerekes, T., et al., Performance of variable flow rates for photovoltaic-thermal collectors and the determination of optimal flow rates. Solar Energy, 2019. 182: p. 148-160.
  2. Hussain, M.I. and J.-T. Kim, Performance evaluation of photovoltaic/thermal (PV/T) system using different design configurations. Sustainability, 2020. 12(22): p. 9520.
  3. Heidarian, A., H. Ghassemi, and P. Liu, Drag reduction by using the microriblet of sawtooth and scalloped types. International Journal of Physics, 2018.
  4. Mahmoodi, K., H. Ghassemi, and A. Heydarian, Solving the Nonlinear Two-Dimension Wave Equation Using Dual Reciprocity Boundary Element Method. International Journal, 2017. 5(1): p. 19-25.

  1. Authors should more clearly indicate the scientific novelty of the structures and methods used.

Authors: the reviewer comment is valid, and it has been considered.

  1. The authors should justify the choice of software used for research in the work.

Authors: the reviewer comment is valid, and it has been considered. We added para;

“In the present simulation, the commercial CFD software ANSYS 17.0 FLUENT software was used, this software is a general-purpose CFD software used to model fluid flow, heat and mass transfer, chemical reactions, and more. Fluent offers a modern, user-friendly interface that streamlines the CFD process from pre- to post-processing within a single window workflow.”

  1. What justified the choice of geometry and the distribution of ledges? Could their other forms and distributions be more effective? It would be advisable to carry out the experiment also with a completely different geometry - for example, a long flat rib, flow turbulators, etc., since the differences between the considered configurations are not so significant.

Authors: the reviewer comment is valid, and it has been considered. We added his suggestions in conclusions as future works.

  1. How and with the help of what did the ribs attach to the metal sheet - what is the thickness and thermal conductivity of the connecting layer? Is a metal sheet added to a standard PV module? What is the thermal conductivity between these layers - after all, standard photovoltaic modules are not designed for hybrid operation. The polysloxane compound used to seal photovoltaic converters serves to increase the optical efficiency and better electrical efficiency of photovoltaic converters when they work with coolants (for example, DOI: 10.4018 / IJEOE. 2020040106) - it is advisable for the authors to keep in mind this manufacturing technology, which is relevant in photovoltaic thermal modules.

Authors: the reviewer comment is valid, and it has been considered.

“During the manufacturing processes, we relied on fixing the fins directly with the bottom face of the collector cover using small screws, and in order to transfer heat, an interfacial coolant was used that increases the thermal conductivity between the surface of the Pv panel and the collector”

  1. Authors should add drawings or sketches of the used module designs with the principle of their operation.

Authors: the reviewer comment is valid, and it has been considered.

We added (a- Steps of the use module designs with the principle of their manufacturing.) in Figure 1.

  1. Judging by Figure 7, the maximum heating of the coolant is only 5 degrees? Then it makes sense to significantly improve the design of the module and its thermal insulation.

Authors: the reviewer comment is valid, and it has been considered.

  1. What is the optimal coolant flow rate for the developed modules? What is the overall efficiency of the developed modules? The developed modules have an electrical efficiency 23.9% and 25.3% (line 402)?

Authors: Thank you on this comment. (line 402) We mean the thermal performance of PV panel (thermal efficiency) increase to 23.9% and 25.3% for both models respectively, at 1:00 pm.

  1. Authors should indicate how and by what formulas the thermal efficiency of modules was calculated.

Authors: the reviewer comment is valid, and it has been considered.

Thermal efficiency ηt

  1. How were the results obtained in Figures 12 and 14? Areas of overheating and stagnation are visible, which indicates the need for further optimization of the module heatsink designs (for example, DOI: 10.4018/978-1-5225-3867-7.ch004), which should be carried out in further research by the authors.

Authors: The comment is really appreciated. This added value to the work. We have used the simulation attempts carried out in the different times at day and data collection of simulated for difference water temperature between the inlet and output. Finally we get results in Figures 12 and 14.

  1. Authors should pay attention to writing English in the work

Authors: the reviewer comment is valid, and it has been considered.

  1. The authors should add a section "Directions for further research", where they indicate the planned further work in this direction. For the modules under consideration, it is advisable to create special solar photovoltaic thermal modules with a pre-developed manufacturing technology and special sealing components of photovoltaic converters, as well as more versatile optimization of the design of the photovoltaic thermal module radiator.

Authors: the reviewer comment is valid, and it has been considered. We added new section

  1. Directions for Further Research

Generally, high operating temperatures cause the shorten life cycle of the PV module due to damaging the module material. This situation has been substantially eliminated by cooling the PV cell. For this reason, prolonged payback time of the PV system, shortening the life of the materials used in PV modules, are among the causes of occurrence high temperature. As a result of the foregoing, the research takes two paths to develop the PV/T system;

1-      Developing a PV/T system by searching for the optimal design of the collector with high thermal efficiency by using methods that increase heat transfer, such as fins or the use of porous materials.

2-      Developing a PV/T system by improving the enhancement thermal fluid properties by using Nano additives.

  1. Also, the authors should add a paragraph at the end of the article indicating the contribution of each author to the content of the work.

Authors: Done. This Para was added.

Contributions

Authors would like to report on the contribution of each author to the content of the work; Mohammed G. Ajel and Basam A. Shallal -They contributed to the experimental part in-cluding the fabrication and setup of the PV/T system and data collection as well as the theoretical part represented by numerical modeling. While Engin Gedik and Hasanain A. Abdul Wahhab, they help with numerical analysis, results ana

Round 2

Reviewer 1 Report

It can be published in the current version.

Reviewer 2 Report

I have no further elements to contribute in the revision of your manuscript.

But I would like to emphasize that the comparison with similar investigations could show the potential of the information presented in this manuscript.

Reviewer 3 Report

The authors have done some work to correct and supplement the article and it can be accepted for publication, but the authors should impeccably add all the indicated answers to the reviewer's comments to the text of the article.